# Synthesis of Ammonium-Based Ionic Liquids for the Extraction Process of a Natural Pigment (Betanin)

**DOI:** 10.3390/molecules26185458

**Published:** 2021-09-08

**Authors:** Pedro Morales-García, Evelyn Y. Calvillo-Muñoz, Irina V. Lijanova, Natalya V. Likhanova, Octavio Olivares-Xometl, Paulina Arellanes-Lozada

**Affiliations:** 1Instituto Politécnico Nacional, CIITEC, Cerrada Cecati S/N, Colonia Santa Catarina de Azcapotzalco, Ciudad de Mexico 02250, Mexico; pmoralesg1201@alumno.ipn.mx (P.M.-G.); ecalvillom1700@alumno.ipn.mx (E.Y.C.-M.); 2Instituto Mexicano del Petróleo, Programa de Investigación y Posgrado, Eje Central Norte Lázaro Cárdenas No. 152, Col. San Bartolo Atepehuacan, Ciudad de Mexico 07730, Mexico; nvictoro@imp.mx; 3Facultad de Ingeniería Química, Benemérita Universidad Autónoma de Puebla, Av. San Claudio y 18 Sur, Col. San Manuel, Ciudad Universitaria, Puebla 72570, Mexico; octavio.olivares@correo.buap.mx (O.O.-X.); paulina.arellanes@correo.buap.mx (P.A.-L.)

**Keywords:** natural beet juice, betanin, extraction process, ammonium-based ionic liquids, kosmotropic salts

## Abstract

The use of new synthesized ammonium-based ionic liquids was explored as an alternative to the current process implemented in the betanin extraction from red beet juice, resulting in high yields: 70% and 82%. Betanin is a vegetal pigment that has been applied to a large variety of products in the food industry, which is important, for it can work as a substitute for the red synthetic dyes used nowadays. Additionally, the use of the kosmotropic salt sodium acetate was explored in order to separate the complex formed by the ionic liquid and pigment of interest in a process that combined two techniques: ATPS (aqueous two-phase system) and SOES (salting-out extraction system). The results reveal that the studied techniques could work as a novel process for the extraction of betanin from red beet juice employing ionic liquids, which have not been tested for this purpose in other research.

## 1. Introduction

Ionic liquids (ILs) have started to play an important role as design-solvents in extraction processes of natural substances: phycobiliproteins from spirulina algae [1], the pigment curcumin [2], and anthocyanins [3]. In most cases, imidazole-based ILs have been widely used because of their high stability, conductivity, miscibility with different substances, and synthesis ease, but unfortunately, at the same time, they tend to have some kind of cytotoxicity for microorganisms [4,5]. In contrast, ammonium-based ILs have shown fewer cytotoxicity effects, so the use of this kind of compound is promising in the food industry [6].

The conditions of extraction processes of natural products are very important because they are susceptible to degradation due to factors such as high temperature (>70 °C), acid or alkaline pH, and air, specifically betanins [7,8]. In this context, there are multiple reports on betanin extraction, employing different techniques: by low DC or pulsed electrical field [9,10,11], g-irradiation [12,13], diffusion apparatus [14], and ultrafiltration and osmotic distillation [15], but extraction by solvents is still the most used due to its simplicity and accessibility.

Aqueous two-phase systems (ATPSs) [16] occur when combinations of hydrophilic solutes (polymers or polymer and certain salts) are incompatible in a specific aqueous solution above critical concentrations, and these systems have proved to be particularly useful in different extraction processes of natural pigments [17,18,19]. The use of ammonium-based ILs and kosmotropic salts in biphasic systems for absorption and desorption processes suggests a new approach for the extraction process of betanins, probably increasing the extraction yields of natural pigments and reducing concomitant problems. Finally, the aims and purpose of this work were the design of a recovery system of betanins from Beta vulgaris (beetroot) using ionic liquids and kosmotropic salts at ambient temperature and atmospheric pressure.

## 2. Results and Discussion

### 2.1. Synthesis of Ammonium-Based ILs

For the design of the ILs used in the present work, it was necessary to consider the possible union sites in the betanin molecule (Figure 1), where the ILs could set up interactions, and to consider the physical properties of the pigment. Analyzing the pigment structure, it can be seen that it has a large number of hydroxyl and carboxyl groups; furthermore, betanins are highly water-soluble pigments, which means that they have a great affinity for aqueous phases.

Additionally, ammonium derivatives display fewer eco- and cytotoxic features than imidazolium-based ILs [5], and the presence of the biodegradable carboxylic anionic part (such as, tartaric or fumaric acids) has been widely exploited by the food and pharmaceutical industries, which is in contrast with the halogen anions present in other ILs [20]. Since it has been reported that the cationic part of ILs is responsible for their physical properties, it was decided to use trihexylamine and trioctylamine to synthesize the IL precursor during the quaternization process with dimethyl carbonate; considering that both amines are hydrophobic compounds with high viscosity, it was suggested that the final products would have high viscosity and low water solubility, which would lead to capture of the pigment in the hydrophobic phase. Four acids were selected to form the anionic part of the extracting ILs, where three of them were dicarboxylic acids (adipic, fumaric, and tartaric) and the fourth was monohydroxybenzoic acid (salicylic) (Table 1).

These compounds were selected because they have similar groups to those found in the betanin structure. A ninth IL was designed using tributylamine during the quaternization process and hexanoic acid to produce the final IL; this new IL had the purpose of decreasing the viscosity of the other eight ILs. Finally, around eight ILs (each one) with two or more carboxylic and/or hydroxyl functional groups in their structures were formed with tributylmethylammoniumhexanoate [TBMA]^+^[HEX]^−^, thus becoming the eight different synergistic extractive system.

### 2.2. Selection of the Best IL System to Absorb the Molecular Structure of Betanin from an Aqueous Solution

From a qualitative point of view, it was observed that the best affinity was displayed by those extracting ILs consisting of trioctylammonium [TOMA]^+^ in their cationic part, indicating (by intensive coloration) that part of the pigment could be retained in the extractor (upper) phase. The specificity of the ILs for the molecular structure of betanin was determined by means of a qualitative extraction test, from which the remaining extracts were subsequently recovered to quantify the amount of collected pigment based on the extraction percentage (P_ex_) (Figure 2).

Based on qualitative and quantitative extraction tests of a commercial pigment (CBE), the best results were shown by the cation [TOMA]^+^, and this effect could be attributed to the increasing number of aliphatic chains, which confirmed the statement that, in general, the physical properties of ILs are given to a greater extent by the cationic part [21]. The ILs [THMA]^+^[ADI]^−^ and [THMA]^+^[FUM]^−^ had similar performance; however, the [TOMA]^+^ structure in the cationic part, particularly [TOMA]^+^[ADI]^−^, reached 68.90% for the commercial pigment. The best extraction system, formed by [TOMA]^+^[ADI]^−^/[TBMA]^+^[HEX]^−^, was used to extract betanin directly from fresh beet juice, where the concentration (309 ppm) of natural pigment was around 3 times greater than the concentration of the commercial product in the same volume (3 mL) (Figure 3).

It should be noted that the extraction process of betanin from a natural source did not pass through pre-treatment; consequently, the aqueous solution could contain different substances, such as sugars, proteins, and amino acids, including some bio-active compounds, such as polyphenols, flavonoids, and glycosides, etc., and these kinds of molecules were capable of interacting with the extracting phase, decreasing the recovery percentage of the pigment of interest [22]. The extraction percentage (P_ex_) results of betanin from beet juice show that they suffered the most significant changes when the extracting complex was added between 0.30 and 1.2 mmol. At these concentrations, the efficiencies remained stable (see the almost linear curve), which was probably due to the formation of a stability state, where the mass transfer was not remarkably increased. The calculations of the distribution coefficient forming the biphasic system ILs/Water (K_IL’s/W_) confirmed the results.

### 2.3. Calculation of Distribution Coefficients in the Biphasic Systems (KIL’s/W)

The distribution coefficients were calculated by different mathematical models—Betageri et al. (1987) [23] and Lima et al. (2017) [3]—obtaining similar results. It is worth mentioning that the obtained distribution coefficients were above 1.0, so betanins had the best affinity with the organic (upper) phase, with 11.24 units of the KIL’s/W. According to the calculations, it is possible to predict that amounts above 0.60 mmol of extraction system are not useful (Figure 4) (Appendix A).

As mentioned before, the extraction processes were carried out in one step, and in order to increase the recovery percentage, it was decided to apply a multi-step approach using 0.30 mmol of IL twice. For this purpose, in the first step, the same volume (3 mL) of beet juice (309 ppm) was put in contact with 0.30 mmol of the extractive phase; then, the recovered refining solution was treated again with the other quantity of complex IL solution. The amounts of recovered pigment after each step were used to calculate the total extraction percentages (P^t^_ex_). Table 2 shows the extraction percentages of betanins (gradually), where during the first step, the extraction parameters of pigment using the additional 0.30 mmol during the second step were better—that is, 0.60 mmol of IL in total. After analyzing the pigment recovery during the next evaluations with additional quantities of IL complex, it was found that in each step, the use of 0.60 mmol (1.20 mmol in total) was excessive. The percentage of total recovery of betanins was significantly increased by applying the multi-step system, where some parameters reached 72% and 82% with 0.30 and 0.60 mmol per step, respectively, in comparison with the one-step system (50% on average).

In contrast, the new calculations of distribution coefficients taking into account the multi-step recovery systems did not show remarkable changes (Figure 5).

The distribution results calculated during the first step were close to the initial ones in Figure 4: 11.61 and 7.84 units, correspondingly, but in the second step, the distribution coefficient decreased to 11.05 and 4.52 units, showing the biggest difference when using 0.60 mmol of IL per step. It signaled that the best multi-step recovery system was when 0.30 mmol of IL was used during in each step.

### 2.4. Aqueous-Based Biphasic System (ILs and Kosmotropic Salts) in the Desorption Process

The extractive systems based on ILs showed promising results during the recovery process of the natural pigment from the aqueous solution; however, another difficult challenge was faced: the desorption process, when betanin has to be separated from the ILs. The interactions between polar organic structures (ILs) and betanin are supposedly based on intermolecular bonds: ion–dipole, H bonds, etc. The successful separation of the ILs and pigment could confirm the nature of their interaction. Some kosmotropic salts (K_3_PO_4_, CH_3_COONa, etc.) have been used during the recovery process of natural pigments—for example, K_3_PO_4_ was applied in an aqueous-based biphasic system (ATPS) to extract anthocyanins [3]; notwithstanding, this kind of salt elevated the pH of the aqueous solution up to 11.5, and since betanins belong to an unstable group of chromophore-molecules, the C=N bond was broken and as a consequence, the pigment suffered the degradation process [24]. The main quality of kosmotropic salts is their capacity to increase the hydrophobicity of some substances, leading to a salting-out effect that favors the stable aggregation state of hydrophobic molecules [19].

Since the use of K_3_PO_4_ represents a disadvantage, for it modifies the solution pH, in the present research work, it was decided to use ethanol and different solutions of sodium acetate at the following concentrations: 5.0%, 10.0%, 15.0%, and 20.0% p/v; however, some authors reported using maximal values of CH_3_COONa until 40% p/v. According to Ball et al. (2015) [19], the counterions could be capable of interacting with hydrophobic molecules (in this case ILs) and at same time of breaking down the intermolecular bonds between the organic phase (ILs) and pigment molecules, releasing the latter. ILs are a stable set with a partial quantity of dissociated salt in the upper phase, while the rest of the counterions form simultaneously ion–dipole interactions or a solvation effect with the water–ethanol phase, displacing the substance with hydrophilic character (betanins). The results point out that the best performance separating betanin (98%) was reached with 20% of sodium acetate in the ethanol solution, confirming the statement that the higher the salt concentration, the better the separation is carried out; but at the same time, the salt could display some different behavior patterns—chaotropic or kosmotropic, depending on its concentration [25].

During the desorption process, two different phases were formed, where the upper phase consisted of the ILs (because their density is lower than that of water), some salt ions, and ethanol traces; the bottom phase was represented by water, natural pigment, some ions, and ethanol. Probably, the ions of the kosmotropic salt promoted an increase in the packaging of the IL hydrophobic aliphatic chains (Van der Waals forces) [25], and at the same time, pushed the betanin hydrophilic molecules to the phase with higher density (water/ethanol), presenting the salting-out effect [26]. In addition, the ion–dipole interactions formed between ions and water and H-bonds formed between water and ethanol, exerting an effect on the separation process (Figure 6).

The dissociation of ions of sodium acetate forces the formation of ion–dipole intermolecular bonds between pigment molecules and salt ions and, at the same time, forces the H-bonds between water molecules and betanin hydroxyl groups. In contrast, the sodium cation could exert a retardant effect on the formation of the transition and the breaking of H-bonds [27], and the synergistic action of the kosmotropic salt/biphasic system led to the successful separation of betanin.

Finally, to confirm the integrity of the betanin molecular structure, a UV analysis was carried out, where the three UV spectra were compared (Figure 7). As can be seen, all the UV-Vis spectra present the characteristic peak of the betanin chromophore group between 530 and 540 nm; in the case of the natural pigment (537 nm) and betanin spectra, after recovery (539 nm), they present a minor bathochromic displacement (2 nm), which is probably due to the interaction with ethanol (405 nm), but in general, the integrity of the betanin molecular structure is confirmed.

## 3. Materials and Methods

### 3.1. General

Tributylamine (99%), trihexylamine (99%), trioctylamine (99%), dimethyl carbonate (99%), adipic acid (99%), fumaric acid (99%), tartaric acid (99%), salicylic acid (99%), hexanoic acid (99%), sodium acetate (99%), heptane (98%), ethanol (99%), and methanol (99%) were purchased from Sigma Aldrich and used as received. Deionized water was acquired from Laboratorio Ambiental CIITEC-IPN; commercial beetroot extract with 12% of betanins (CBE) and beetroot were bought at a local market.

All the synthesized compounds were characterized by ^1^H and ^13^C NMR and infrared (IR) and ultraviolet–visible (UV-Vis) spectroscopic techniques. IR/UV-Vis spectra were recorded on a Thermo Scientific Nicolet 8700 spectrometer. ^1^H (300 MHz) and ^13^C NMR (75.40 MHz) spectra were recorded on a JEOL Eclipse-300 in CDCl_3_, and chemical shifts were expressed in ppm with tetramethylsilane as the internal standard.

### 3.2. IL synthesis and Characterization

The process was carried out in a Parr^®^ 4848 (special design) reactor at 160 °C, where a tertiary amine (tributylamine, trihexylamine, trioctylamine) reacted with dimethyl carbonate with a 1:2 molar ratio using methanol as the reaction medium. The final products were washed with 40 mL of heptane: a methanol mixture with 1:1 volumetric ratio to eliminate the rest of the tertiary amine that did not react and generated sub-products. Then, the anionic exchange was performed in 100 mL flasks by mixing equimolar quantities of the IL precursor and adipic, fumaric, tartaric, and salicylic acids, respectively; also using 40 mL of ethanol as reaction medium and stirring at 40 °C for 30 min. In all the cases, the ILs were dried under vacuum employing two pieces of equipment: first, a Rotavapor^®^ Buchi R-215, and second, a Vacío^®^ EDV-A300. This process helped eliminate the remaining solvents. The final ILs produced after the exchange process are reported in Table 1.

*Tributylmethylammoniummethylcarbonate* [TBMA]^+^[MCA]^−^: amber viscous liquid, 75% yield.

FTIR/ATR (cm^−1^): 3218, 2961, 2869, 2667, 1644, 1461, 1369, 1286, 1069, 878, 744, 686. ^1^H NMR (301 MHz, CDCl_3_), δH (ppm): 1.00 (t, 9H, CH_3_), 1.44 (m, 6H, CH_2_), 1.67 (br, 6H, CH_2_), 3.18 (s, 3H, CH_3_-N), 3.32 (br, 3H, CH_2_-N), 3.37 (s,6H, CH_3_-O). ^13^C NMR (76 MHz, CDCl_3_) δC (ppm): 13.68 (CH_3_), 19.67 (CH_2_), 24.22 (CH_2_), 49.55 (CH_3_-N), 52.29 (CH_2_-N), 61.23 (CH_3_-O), 158.58 (C=O).

*Trihexylmethylammoniummethylcarbonate* [THMA]^+^[MCA]^−^: amber viscous liquid, 67% yield.

FTIR/ATR (cm^−1^): 3402, 2937, 2860, 2653, 1620, 1466, 1379, 992, 828, 691. ^1^H NMR (301 MHz, CDCl_3_) δH (ppm): 0.89 (t, 9H, CH_3_), 1.34 (br, 18H, CH_2_), 1.66 (br, 6H, CH_2_), 3.15 (s, 3H, CH_3_-N), 3.25 (br, 6H, CH_2_-N), 3.39 (s, 3H, CH_3_-O). ^13^C NMR (76 MHz, CDCl_3_) δC (ppm): 13.95 (CH_3_), 22.15 (CH_2_), 22.41 (CH_2_), 25.89 (CH_2_), 31.22 (CH_2_), 48.95 (CH_3_-N), 57.21 (CH_2_-N), 61.35 (CH_3_-O), 160.47 (C=O).

*Trioctylmethylammoniummethylcarbonate* [TOMA]^+^[MCA]^−^: amber viscous liquid, 65% yield.

FTIR/ATR (cm^−1^): 3225, 2924, 2854, 1659, 1464, 1278, 1073, 880, 720. ^1^H NMR (301 MHz, CDCl_3_) δH (ppm): 0.88 (t, 9H, CH_3_), 1.27 (br, 30H, CH_2_), 1.66 (br, 6H, CH_2_), 3.19 (s, 3H, CH_3_-N), 3.31 (br, 6H), 3.38 (s, 3H, CH_3_-O). ^13^C NMR (76 MHz, CDCl_3_) δC (ppm): 14.09 (CH_3_), 22.33 (CH_2_), 22.62 (CH_2_), 26.33 (CH_2_), 29.08 (CH_2_), 29.16 (CH_2_), 31.70 (CH_2_), 49.64 (CH_3_-N), 52.40 (CH_2_-N), 61.37 (CH_3_-O), 158.70 (C=O).

*Trihexylmethylammoniumadipate* [THMA]^+^[ADI]^−^: amber viscous liquid, 85% yield.

FTIR/ATR (cm^−1^): 3417, 2930, 2861, 2500, 1943, 1712, 1561, 1464, 1374, 1207, 880. ^1^H NMR (301 MHz, CDCl_3_) δH (ppm): 0.89 (t, 9H, CH_3_), 1.33 (s, 18H, CH_2_), 1.61 (br, 10H, CH_2_), 2.24 (t, 4H, CH_2_), 3.12 (s, 3H, CH_3_-N), 3.25 (br, 6H, CH_2_-N). ^13^C NMR (75 MHz, CDCl_3_) δC (ppm): 13.93 (CH_3_), 22.17 (CH_2_), 22.41 (CH_2_), 25.68 (CH_2_), 25.93 (CH_2_), 31.21 (CH_2_), 36.18 (CH_2_), 48.86 (CH_3_-N), 61.44 (CH_2_-N), 178.37 (COO^−^).

*Trioctylmethylammoniumadipate* [TOMA]^+^[ADI]^−^: amber viscous liquid, 89% yield.

FTIR/ATR (cm^−1^): 3385, 2924, 2863, 1943, 2492, 1703, 1570, 1464, 1455, 1385, 1207, 880. ^1^H NMR (301 MHz, CDCl_3_) δH (ppm): 0.88 (t, 9H, CH_3_), 1.27 (br, 30H, CH_2_), 1.64 (br, 10H, CH_2_), 2.26 (t, 4H, CH_2_), 3.14 (s, 3H, CH_3_-N), 3.26 (br, 6H, CH_2_-N). ^13^C NMR (76 MHz, CDCl_3_) δC (ppm): 14.09 (CH_3_), 22.24 (CH_2_), 22.60 (CH_2_), 25.72 (CH_2_), 26.29 (CH_2_), 29.05 (CH_2_), 29.09 (CH_2_), 31.67 (CH_2_), 36.20 (CH_2_), 48.85 (CH_3_-N), 61.42 (CH_2_-N), 178.19 (COO^−^).

*Trihexylmethylammoniumfumarate* [THMA]^+^[FUM]^−^: amber viscous liquid, 84% yield.

FTIR/ATR (cm^−1^): 3412, 2933, 2863, 2474, 1888, 1703, 1464, 1374, 1242, 985. ^1^H NMR (301 MHz, CDCl_3_) δH (ppm): 0.87 (t, 9H, CH_3_), 1.31 (s, 18H, CH_2_), 1.66 (br, 6H, CH_2_), 3.12 (s, 3H, CH_3_-N), 3.26 (br, 6H, CH_2_-N), 6.75 (s, 2H, CH=CH). ^13^C NMR (76 MHz, CDCl_3_) δC (ppm): 13.79 (CH_3_), 21.98 (CH_2_), 22.21 (CH_2_), 25.72 (CH_2_), 30.98 (CH_2_), 48.52 (CH_3_-N), 61.33 (CH_2_-N), 135.33 (CH=CH), 169.65 (COO^−^).

*Trioctylmethylammoniumfumarate* [TOMA]^+^[FUM]^−^: amber viscous liquid, 86% yield.

FTIR/ATR (cm^−1^): 3368, 2924, 2854, 2500, 1906, 1703, 1570, 1464, 1374, 1242, 985. ^1^H NMR (301 MHz, CDCl_3_) δH (ppm): 0.87 (t, 9H, CH_3_), 1.25 (br, 30H, CH_2_), 1.65 (br, 6H, CH_2_), 3.15 (s, 3H, CH_3_-N), 3.27 (br, 6H, CH_2_-N), 6.77 (s, 2H, CH=CH). ^13^C NMR (76 MHz, CDCl_3_) δC (ppm): 14.12 (CH_3_), 22.22 (CH_2_), 22.61 (CH_2_), 26.23 (CH_2_), 26.93 (CH_2_), 29.02 (CH_2_), 31.68 (CH_2_), 48.84 (CH_3_-N), 61.45 (CH_2_-N), 135.49 (CH=CH), 170.14 (COO^−^).

*Trihexylmethylammoniumsalicylate* [THMA]^+^[SAL]^−^: amber viscous liquid, 85% yield.

FTIR/ATR (cm^−1^): 3412, 2933, 2854, 1925, 1888, 1802, 1631, 1589, 1496, 1137, 1029, 762, 721. ^1^H NMR (301 MHz, CDCl_3_) δH (ppm): 0.85 (t, 9H, CH_3_), 1.22 (s, 18H, CH_2_), 1.50 (br, 6H, CH_2_), 3.00 (s, 3H, CH_3_-N), 3.08 (br, 6H, CH_2_-N), 6.72 (t, 1H, -CH=), 6.81 (d, 1H, -CH=), 7.23 (t, 1H, -CH=), 7.91 (d, 1H, -CH=). ^13^C NMR (76 MHz, CDCl_3_) δC (ppm): 13.87 (CH_3_), 22.15 (CH_2_), 22.36 (CH_2_), 25.83 (CH_2_), 31.13 (CH_2_), 48.60 (CH_3_-N), 61.52 (CH_2_-N), 116.36 (-CH=), 117.46 (-CH=), 119.25 (-C=), 130.65 (-CH=), 132.53 (-CH=), 162.13 (OH-CH=), 173.68 (COO^−^).

*Trioctylmethylammoniumsalicylate* [TOMA]^+^[SAL]^−^: amber viscous liquid, 88% yield.

FTIR/ATR (cm^−1^): 3403, 2924, 2854, 1925, 1897, 1802, 1631, 1587, 1494, 1137, 1021, 762, 721. ^1^H NMR (301 MHz, CDCl_3_) δH (ppm): 0.87 (t, 9H, CH_3_), 1.22 (s, 30H, CH_2_), 1.51 (br, 6H, CH_2_), 3.01 (s, 3H, CH_3_-N), 3.11 (br, 6H, CH_2_-N), 6.71 (t, 1H, (-CH=)), 6.81 (d, 1H, (-CH=)), 7.22 (t, 1H, (-CH=)), 7.91 (d, 1H, (-CH=)). ^13^C NMR (76 MHz, CDCl_3_) δC (ppm): 14.05 (CH_3_), 22.19 (CH_2_), 22.56 (CH_2_), 26.15 (CH_2_), 26.85 (CH_2_), 28.97 (CH_2_), 31.60 (CH_2_), 48.51 (CH_3_-N), 61.42 (CH_2_-N), 116.27 (-CH=), 117.25 (-CH=), 119.57 (-C=), 130.59 (-CH=), 132.27 (CH=), 162.21 (OH-CH=), 173.69 (COO^−^).

*Trihexylmethylammoniumtartarate* [THMA]^+^[TAR]^−^: amber viscous liquid, 84% yield.

FTIR/ATR (cm^−1^): 3421, 2933, 2854, 2492, 1721, 1464, 1120, 1109, 889. ^1^H NMR (301 MHz, CDCl_3_) δH (ppm): 0.89 (t, 9H, CH_3_), 1.33 (s, 18H, CH_2_), 1.66 (br, 6H, CH_2_), 3.09 (s, 3H, CH_3_-N), 3.23 (br, 6H, CH_2_-N), 4.28 (s, 2H, CH), 6.71 (br, 3H, C-OH). **^13^C NMR** (76 MHz, CDCl_3_) δC (ppm): 13.90 (CH_3_), 22.15 (CH_2_), 22.38 (CH_2_), 25.89 (CH_2_), 31.14 (CH_2_), 48.86 (CH_3_-N), 61.61 (CH_2_-N), 71.87 (C-OH), 175.54 (COO^−^).

*Trioctylmethylammoniumtartarate* [TOMA]^+^[TAR]^−^: amber viscous liquid, 87% yield.

FTIR/ATR (cm^−1^): 3412, 2933, 2854, 2500, 1721, 1470, 1119, 977, 880. ^1^H NMR (301 MHz, CDCl_3_) δH (ppm): 0.88 (t, 9H, CH_3_), 1.66 (d, 30H, CH_2_), 1.66 (br, 6H, CH_2_), 3.09 (s, 3H, CH_3_-N), 3.24 (br, 6H, CH_2_-N), 4.22 (s, 2H, CH), 6.53 (br, 3H, C-OH). ^13^C NMR (76 MHz, CDCl_3_) δC (ppm): 14.09 (CH_3_), 22.22 (CH_2_), 22.59 (CH_2_), 26.25 (CH_2_), 26.79 (CH_2_), 29.00 (CH_2_), 31.63 (CH_2_), 48.83 (CH_3_-N), 61.65 (CH_2_-N), 71.65 (C-OH), 175.60 (COO^−^).

*Tributylmethylammoniumhexanoate* [TBMA]^+^[HEX]^−^: amber viscous liquid, 84% yield.

FTIR/ATR (cm^−1^): 3394, 2960, 2872, 1640, 1561, 1394, 1065, 897, 746. ^1^H NMR (301 MHz, MeOD) δH (ppm): 0.63 (t, 3H, CH_3_), 0.74 (t, 9H, CH_3_), 1.05 (br, 4H, CH_2_), 1.13 (m, 6H, CH_2_), 1.32 (m, 2H, CH_2_), 1.42 (m, 6H, CH_2_), 1.88 (t, 2H, CH_2_), 2.75 (s, 3H, CH_3_), 3.00 (br, 6H, CH_2_). ^13^C NMR (76 MHz, MeOD) δC (ppm): 14.00 (CH_3_), 14.13 (CH_3_), 18.68 (CH_2_), 20.85 (CH_2_), 21.97 (CH_2_), 25.28 (CH_2_), 29.87 (CH_2_), 58.45 (CH_3_), 62.68 (CH_2_), 161.15 (COO^−^).

### 3.3. Extraction Tests of Banin

This process was carried out mixing equimolar quantities of [TBMA]^+^[HEX]^−^ and another IL obtained after the anionic exchange. In the end, eight extraction IL complexes were obtained and used in the first extraction tests. A 105 mg/L solution was prepared using commercial betanins (12% of pigment), deionized water, and ascorbic acid adjusted at 0.10% (*w/v*) in the final solution (final pH 6). Quantities equivalent to 0.001 and 0.60 mmol of extraction complexes were weighed in 4 mL jars; later, 3 mL of pigment solution was added to each jar. The final system was stirred with a magnetic bar for 1 h at environment pressure and temperature. After the stirring process, the remaining betanin were quantified using Equation:$$C_{f}=D_{F}\left(\frac{Abs-0.0035}{0.1154}\right)$$
which was performed by making a calibration curve at different concentrations of betanins dissolved in water with 0.10% of ascorbic acid (pH 6).

The red beet root was chopped in small 5 mm cubes and frozen at −5 °C. Fifty grams was weighed and mixed with a solution of ascorbic acid at 0.10% (*w*/*v*) in a porcelain mortar.The cubes were crushed until obtaining a fine pulp, which was filtered using a vacuum in order to collect the liquid extract in a flask. By the end of this process, 60 mL of the extract was recovered and assessed by UV-Vis at 538 nm using same Equation, mentioned before calculate the betalaine content in mg/L.

### 3.4. Desorption Process

Once the aqueous phase was removed, the remaining organic phase was treated with 400 µL of ethanol and 700 µL of sodium acetate solution at specific concentrations: 5%, 10%, 15%, and 20% (*w*/*v*) were evaluated for optimization purposes. The sodium acetate solution was enriched with 0.10% of ascorbic acid. The system was stirred for 2 min, and it was later allowed to rest for 20 min to stabilize the phases.

## 4. Conclusions

The system consisting of ammonium ILs [TOMA]^+^[ADI]^−^/[TBMA]^+^[HEX]^−^ was capable of extracting betanin from an aqueous solution with yields above 70% for a commercial pigment and 82% for a natural pigment from fresh beet juice. The best results of the betanin extraction process were achieved using a multi-step recovery approach. In addition, the distribution coefficient was calculated, where the best results were presented at 30 mmol with 11 units. Finally, the desorption process was successfully carried out, using kosmotropic salt (CH_3_COONa—20%), confirming the integrity of the recovered natural pigment (betanin).

## Figures and Tables

**Figure 1 molecules-26-05458-f001:**
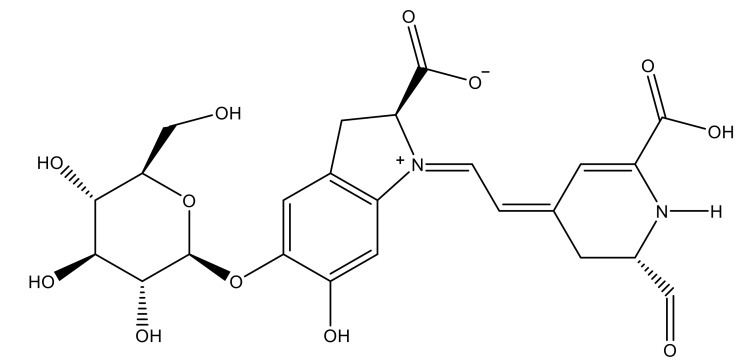
Structure of betanin.

**Figure 2 molecules-26-05458-f002:**
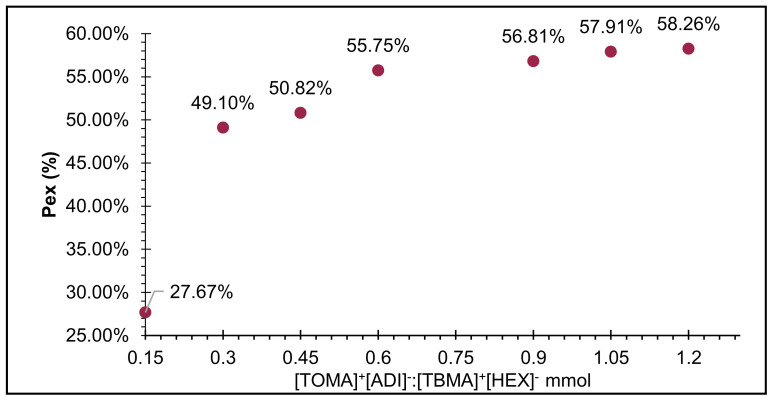
Average results of extraction percentage (P_ex_) of CBE by different systems.

**Figure 3 molecules-26-05458-f003:**
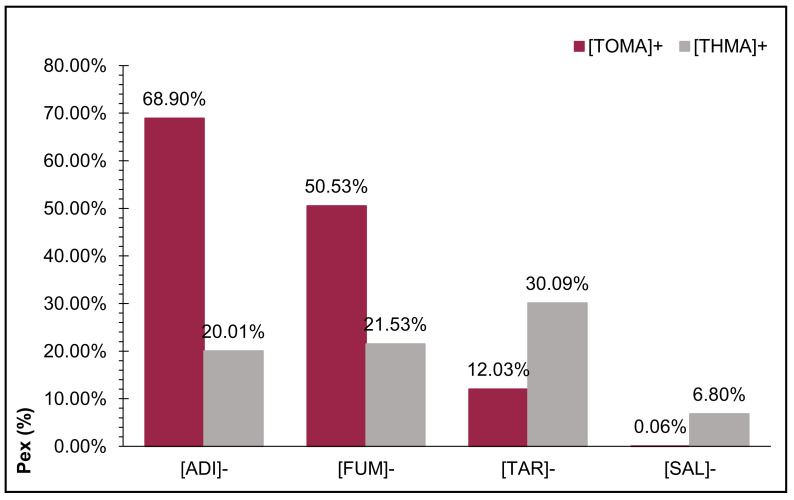
Average results of extraction percentage (P_ex_) of betanin from beet juice by [TOMA]^+^[ADI]^−^/[TBMA]^+^[HEX]^−^ in one step.

**Figure 4 molecules-26-05458-f004:**
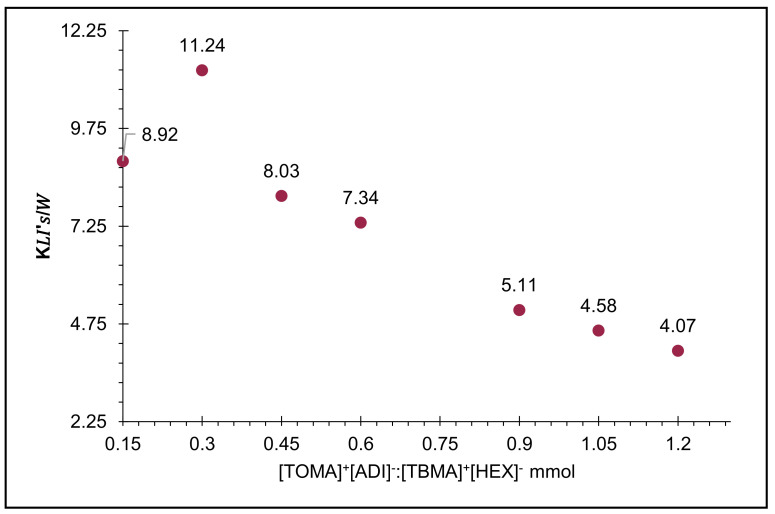
Average results of distribution coefficient calculations of biphasic systems: ILs/water (K_IL’s/W_).

**Figure 5 molecules-26-05458-f005:**
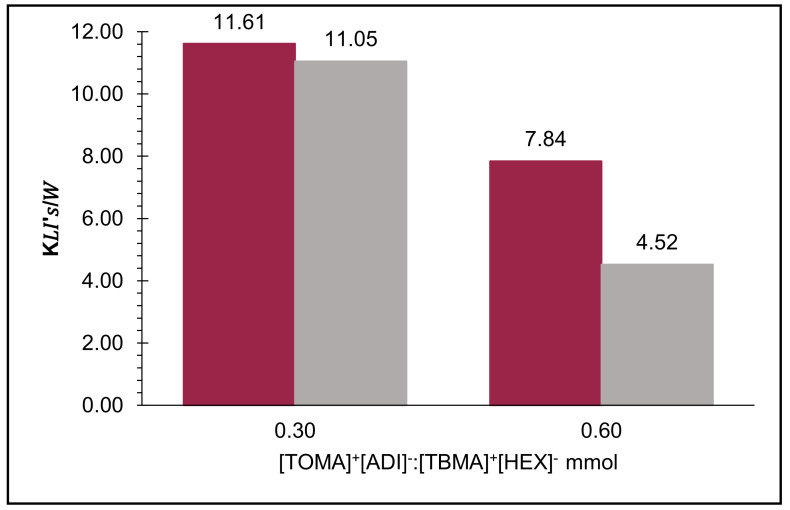
Distribution coefficients after two steps (1st step—red color, 2nd step—grey color).

**Figure 6 molecules-26-05458-f006:**
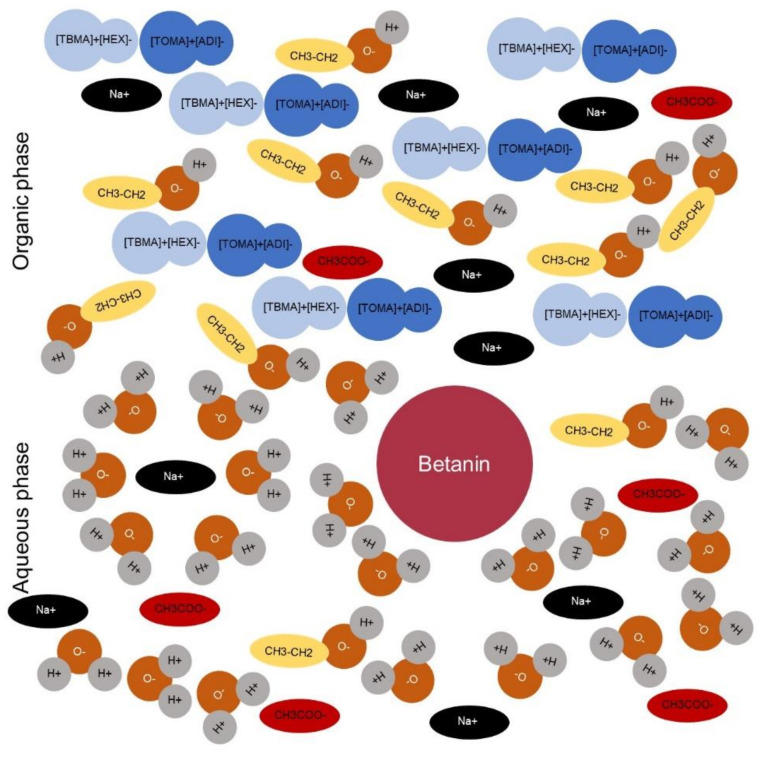
Formation of two phases during the desorption process using the kosmotropic salt.

**Figure 7 molecules-26-05458-f007:**
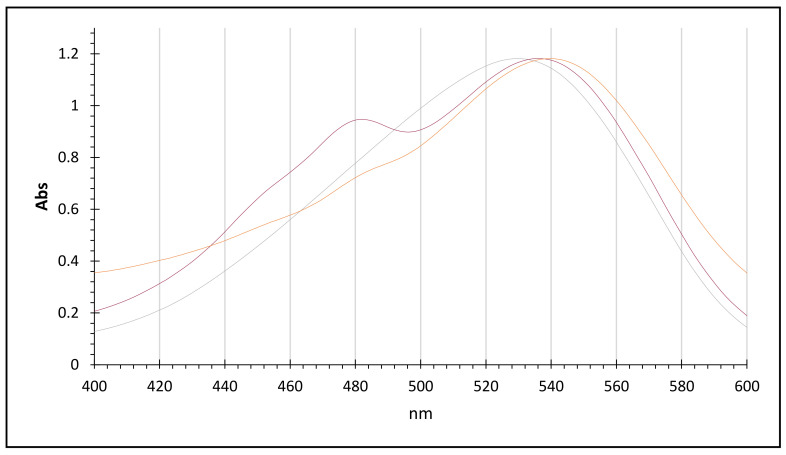
The three UV-Vis spectra of betanin: commercial pigment—grey line, natural pigment (beet juice)—red line, natural pigment after recovery—orange line.

**Table 1 molecules-26-05458-t001:** Structure of the synthesized ILs.

Structure	Name	Abbreviation	Yields
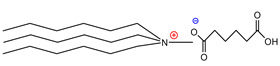	Trihexylmethylammoniumadipate	[THMA]^+^[ADI]^−^	85%
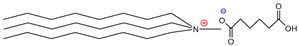	Trioctylmethylammoniumadipate	[TOMA]^+^[ADI]^−^	89%
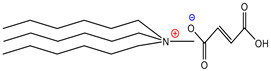	Trihexylmethylammoniumfumarate	[THMA]^+^[FUM]^−^	84%
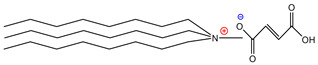	Trioctylmethylammoniumfumarate	[TOMA]^+^[FUM]^−^	86%
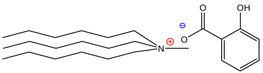	Trihexylmethylammoniumsalicylate	[THMA]^+^[SAL]^−^	85%
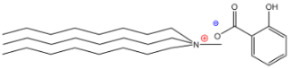	Trioctylmethylammoniumsalicylate	[TOMA]^+^[SAL]^−^	88%
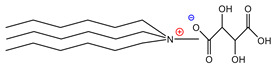	Trihexylmethylammoniumtartarate	[THMA]^+^[TAR]^−^	84%
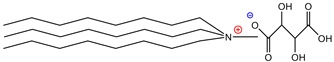	Trioctylmethylammoniumtartarate	[TOMA]^+^[TAR]^−^	87%
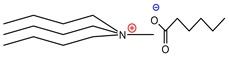	Tributylmethylammoniumhexanoate	[TBMA]^+^[HEX]^−^	84%

**Table 2 molecules-26-05458-t002:** Recovery percentages of betanin (initial 1.40 × 10^−3^ mmol) after two steps.

[TOMA]^+^[ADI]^−^/[TBMA]^+^[HEX]^−^ Per Step (mmol)	Steps	P_ex_	Betanin Recovery (mmol) (10^−3^)	Total, Betanin Recovery (mmol) (10^−3^) P^t^_ex_ (%)
0.30	1	49.91%	0.70	
	2	48.67%	0.36	1.04 74.30%
0.60	1	57.35%	0.60	
	2	43.67%	0.25	1.15 82.40%

## Data Availability

All the data associated is contained within this article.

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
