# Peer review of "Synthesis of Ammonium-Based Ionic Liquids for the Extraction Process of a Natural Pigment (Betanin)"

_molecules, 2021, doi:10.3390/molecules26185458_

Round 1

Reviewer 1 Report

Authors designed and synthesized a serious ILs to be applied in biphasic systems for extracting betanin from commercial pigments and fresh red beet juice, good high yields were achieved. The aqueous two-phase systems with CH3COONa were used in desorption processes, displaying excellent capacity (98%) to recover the pigment from the extractive organic phase. It is recommended to accept for publication after revision.

  1. If CH3COONa is used in desorption process, how to recover the used ILs?
  2. After separation, how much IL remained in the recovered betanins? It is known it is very difficult to recover organic compounds in IL with high boiling points.
  3. Some results should be moved to supporting information, for example, FT-IR spectra, calculations.
  4. The concentration of betanins used in the study is very low, please comment on it.

Reviewer 2 Report

Type:

Article Paper  

The authors reported a full paper: The system consisting of ammonium ILs [TOMA]+[ADI]-/ [TBMA]+[HEX]- was capable of extracting betanin from an aqueous solution.

The paper is good and require some English editing

Topic should be revised to: Synthesis of ammonium-based ionic liquids for the extraction process of a natural pigment (betanin)

Abstract

Abstract should be improved to enhance understanding.

  1. Introduction

Aims and purpose of the study should be included in this section.

  1. Results and Discussion

2.3. Calculation of distribution coefficients in the biphasic systems (KIL's/W).

Betageri, G. V., et al., 1987 [23] and Lima, Á. S., et al., 2017 [24],….is his a proper reference citation pattern?

Table 2. Recovery percentages of betalaine (initial 1.40x10-3 mmol) after two steps….. Betalaine recovery…..correct the word  ‘betalaine’

Line 178… By taking into account the…please, rephrase the starting sentence.

4.3. Extraction tests of betanin

4.2. IL synthesis and characterization

This process should be rephrased and well defined for understanding

Line: 344-345: The final system was stirred with a magnetic bar for 1 h at environment pressure and temperature….What are the environmental pressure and temperature used in the study for reproducibility purpose?

Equation 1: where and how are the factors 0.0035 and 0.1154 obtained for this purpose.

Line 366: …20 % (w/v) were tested to determine which of them could provide the best results…..This sentence can be changed to …were evaluated for optimization purposes.

Funding: Not applicable and Acknowledgments: This paper has been made possible by the 20210901 project of the SIP IPN……these two statements are not in agreement as stated by the authors….Authors should clarify or rephrase.

References

The references are not in accordance with the journal reference style. Please check...

Round 2

Reviewer 1 Report

Authors revised the manuscript as pointed out. It is recommended to accept for publication.

Reviewer 2 Report

Manuscript revised